# Generating High Fidelity Images with Subscale Pixel Networks and Multidimensional Upscaling

**Jacob Menick**[*]
DeepMind
jmenick@google.com

**Nal Kalchbrenner**[*]
Google Brain Amsterdam
nalk@google.com

## Abstract

The unconditional generation of high fidelity images is a longstanding benchmark for testing the performance of image decoders. Autoregressive image models have been able to generate small images unconditionally, but the extension of these methods to large images where fidelity can be more readily assessed has remained an open problem. Among the major challenges are the capacity to encode the vast previous context and the sheer difficulty of learning a distribution that preserves both global semantic coherence and exactness of detail. To address the former challenge, we propose the Subscale Pixel Network (SPN), a conditional decoder architecture that generates an image as a sequence of sub-images of equal size. The SPN compactly captures image-wide spatial dependencies and requires a fraction of the memory and the computation required by other fully autoregressive models. To address the latter challenge, we propose to use Multidimensional Upscaling to grow an image in both size and depth via intermediate stages utilising distinct SPNs. We evaluate SPNs on the unconditional generation of CelebAHQ of size 256 and of ImageNet from size 32 to 256. We achieve state-of-the-art likelihood results in multiple settings, set up new benchmark results in previously unexplored settings and are able to generate very high fidelity large scale samples on the basis of both datasets.

## 1 Introduction

A successful generative model has two core aspects: it produces targets that have high fidelity and it generalizes well on held-out data. Autoregressive (AR) models trained by conventional maximum likelihood estimation (MLE) have produced superior scores on held-out data across a wide range of domains such as text (Vaswani et al., 2017; Wu et al., 2016), audio (van den Oord et al., 2016a), images (Parmar et al., 2018) and videos (Kalchbrenner et al., 2016). These scores are a measure of the models' ability to generalize in that setting. From the perspective of sample fidelity, the outputs generated by AR models have also achieved state-of-the-art fidelity in many of the aforementioned domains with one notable exception. In the domain of unconditional large-scale image generation, AR samples have yet to manifest long-range structure and semantic coherence.

One source of difficulties impeding high-fidelity image generation is the multi-faceted relationship between the MLE scores achieved by a model and the model's sample fidelity. On the one hand, MLE is a well-defined measure as improvements in held-out scores generally produce improvements in the visual fidelity of the samples. On the other hand, as opposed to for example adversarial methods (Arora & Zhang, 2017), MLE forces the model to support the entire empirical distribution. This guarantees the model's ability to generalize at the cost of allotting capacity to parts of the distribution that are irrelevant to fidelity. A second source of difficulties arises from the high dimensionality of large images. A $256 \times 256 \times 3$ image has a total of 196,608 positions that need to be architecturally connected in order to learn dependencies among them; the representations at each position require sufficient capacity to express their respective surrounding contexts. These requirements translate to large amounts of memory and computation.

---

[*]Equal contributions.

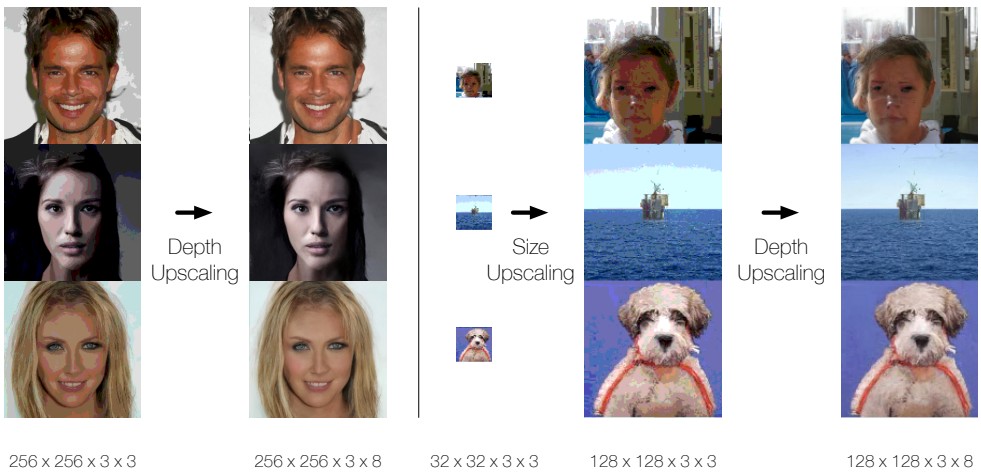

256 x 256 x 3 x 3          256 x 256 x 3 x 8          32 x 32 x 3 x 3          128 x 128 x 3 x 3          128 x 128 x 3 x 8

Figure 1: A representation of Multidimensional Upscaling. Left: depth upscaling is applied to a *generated* 3-bit $256 \times 256$ RGB subimage from CelebAHQ to map it to a full 8-bit $256 \times 256$ RGB image. Right: size upscaling followed by depth upscaling are applied to a *generated* 3-bit $32 \times 32$ RGB subimage from ImageNet to map it to the target resolution of the 8-bit $128 \times 128$ RGB image. We stress that the rightmost column of both figures are true unconditional samples from our model at full 8bit depth.

These difficulties notwithstanding, we aim to learn the full distribution over 8-bit RGB images of size up to $256 \times 256$ well enough so that the samples have high fidelity. We aim to guide the model to focus first on visually more salient bits of the distribution and later on the visually less salient bits. We identify two visually salient subsets of the distribution: first, the subset determined by sub-images ("slices") of smaller size (e.g. $32 \times 32$) sub-sampled at all positions from the original image; and secondly, the subset determined by the few (e.g. 3) most significant bits of each RGB channel in the image. We use *Multidimensional Upscaling* to map from one subset of the distribution to the other one by upscaling images in size or in depth. For example, the generation of a $128 \times 128$ 8-bit RGB image proceeds by first upscaling it in size from a $32 \times 32$ 3-bit RGB image to a $128 \times 128$ 3-bit RGB image; we then upscale the resulting image in depth to the original resolution of the $128 \times 128$ 8-bit RGB image. We thus train three networks: (a) a decoder on the small size, low depth image slices subsampled at every $n$ pixels from the original image with the desired target resolution; (b) a size-upscaling decoder that generates the large size, low depth image conditioned on the small size, low depth image; and (c) a depth-upscaling decoder that generates the large size, high depth image conditioned on the large size, low depth image. Figure 1 illustrates this process.

To address the latter difficulties that ensue in the training of decoders (b) and (c), we develop the Subscale Pixel Network (SPN) architecture. The SPN divides an image of size $N \times N$ into sub-images of size $\frac{N}{S} \times \frac{N}{S}$ sliced out at interleaving positions (see Figure 2), which implicitly also captures a form of size upscaling. The $N \times N$ image is generated one slice at a time conditioned on previously generated slices in a way that encodes a rich spatial structure. SPN consists of two networks, a conditioning network that embeds previous slices and a decoder proper that predicts a single target slice given the context embedding. The decoding part of the SPN acts over image slices with the same spatial structure and it can share weights for all of them. The SPN is an independent image decoder with an implicit size upscaling mechanism, but it can also be used as an *explicit* size upscaling network by initializing the first slice of the SPN input at sampling time with one generated separately during step (a).

We extensively evaluate the performance of SPN and the size and depth upscaling methods both quantitatively and from a fidelity perspective on two unconditional image generation benchmarks, CelebAHQ-256 and ImageNet of various sizes up to 256. From a MLE scores perspective, we compare with previous work to obtain state-of-the-art results on CelebAHQ-256, both at full 8-bit resolution and at the reduced 5-bit resolution (Kingma & Dhariwal, 2018), and on ImageNet-64. We also establish MLE baselines for ImageNet-128 and ImageNet-256. From a sample fidelity

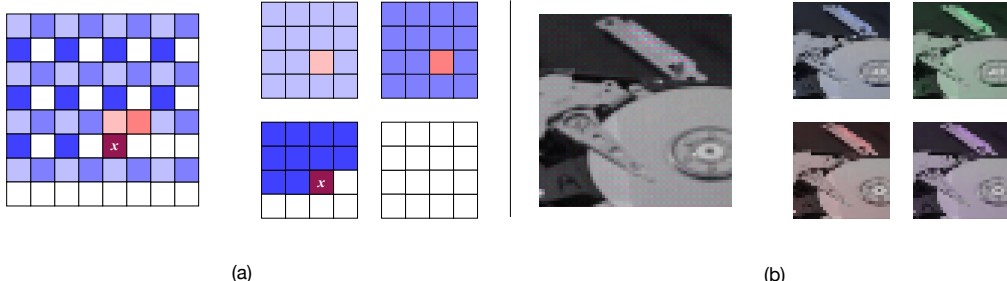

(a)              (b)

Figure 2: The receptive field in a Subscale Pixel Networks (a) and the four image slices subsampled from the original image (b)

perspective, we show the strong benefits of multidimensional upscaling as well as the benefits of the SPN. We produce CelebAHQ-256 samples (at full 8-bit resolution) that are of similar visual fidelity to those produced with methods such as GANs that lack however an intrinsic measure of generalization (Mescheder, 2018; Karras et al., 2017). We also produce some of the first successful samples on unconditional ImageNet-128 (also at 8-bit) showing again the striking impact of the SPN and of multidimensional upscaling on sample quality and setting a fidelity baseline for future methods.

## 2 MODEL

### 2.1 CONVENTIONAL GENERATION ORDERING

A standard AR image model such as the PixelCNN (van den Oord et al., 2016b) generates an $HxW$ colour image starting at the top-left position and ending at the bottom-right position, fully generating the three 8-bit channels of each pixel in a given position:

$$P(\mathbf{x}) = \prod_{h=1}^{H} \prod_{w=1}^{W} \prod_{c}^{\{R,G,B\}} P(x_{h,w,c}|\mathbf{x}_<) \qquad (1)$$

where $\mathbf{x}_<$ corresponds to all previously generated intensity values in the ordering and $h$, $w$, and $c$ are row, column, and colour channel indices. The raster scan ordering (Figure 3(a)) is conventionally used in AR models. Each conditional distribution $P(x_{h,w,c}|\mathbf{x}_<)$ is parametrized by a deep neural network (van den Oord et al., 2016b).

### 2.2 SUBSCALE ORDERING IN IMAGES

We define an alternative ordering that divides a large image into a sequence of equally sized slices and has various core properties. First, it makes it easy to compactly encode long-range dependencies across the many pixels in the large images. It also induces a spatial structure over the original image by aligning the subsampled slices; this also has an implicit size upscaling side effect. From the perspective of the neural architecture, it makes it possible for the same decoder within the SPN to be consistently applied to all slices, since they are structurally similar; the smaller slices also allow for self-attention (Vaswani et al., 2017) in the SPN to be used without local contexts (Parmar et al., 2018). We think of the ordering as the two-dimensional analogue of the one-dimensional subscale ordering introduced in Kalchbrenner et al. (2018).

The subscale ordering is defined as follows:

$$P(\mathbf{x}) = \prod_{i=1}^{S} \prod_{j=1}^{S} \prod_{h=1}^{H/S} \prod_{w=1}^{W/S} \prod_{c}^{\{R,G,B\}} P(x_{i+S*h,j+S*w,c}|\mathbf{x}_<) \qquad (2)$$

where $\mathbf{x}_<$ corresponds to all previously generated intensity values according to this ordering. Figure 3(d) illustrates the subscale ordering. A scaling factor $S$ is selected and each slice of size $H/S \times W/S$ is obtained by selecting a pixel every $S$ pixels in both height and width; there are thus $S^2$ interleaved slices in the original image, with each specified by its row and column offset $(i, j)$. We sometimes refer to this offset as the "meta-position" of a slice.

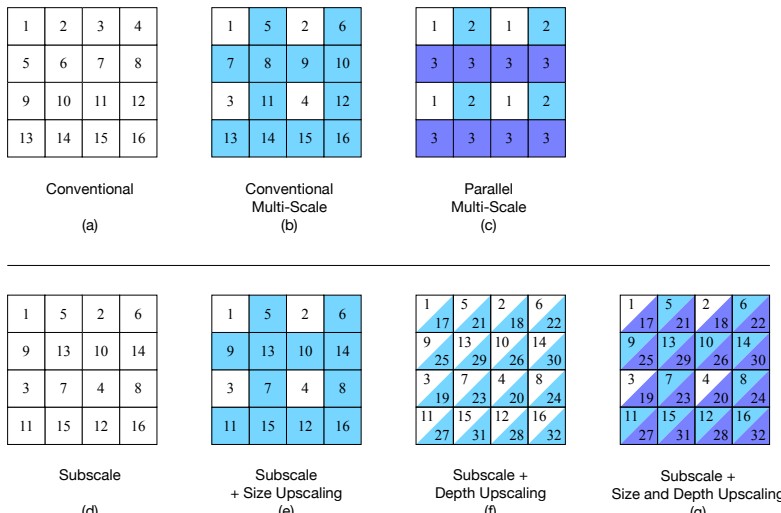

Figure 3: Different generation ordering schemes, where the numbers indicate the step-by-step order. Distinct colors correspond to distinct neural networks. (a) and (b) are from (van den Oord et al., 2016b). (c) is from (Reed et al., 2017). The Subscale ordering alone, with size-only, depth-only and with multidimensional upscaling are, respectively, in blocks (d), (e), (f) and (g).

## 2.3 Size Upscaling in Subscale Ordering

The subscale ordering itself already captures size upscaling implicitly. Analogous to the multi-scale ordering (van den Oord et al., 2016b), and depicted in 3(b), we can perform size upscaling *explicitly*, by training a single slice decoder on subimages and generate the first slice of a subscale ordering from the single slice decoder itself. The rest of the image is then generated according to the subscale ordering by the main network (see 3(e)). The single-slice model can be trained on just the first slices of images, or on slices at all positions in all images given the shared spatial structure among the slices. For this reason, the same SPN that captures the subscale ordering can act simultaneously as a full-blown image model as well as a size upscaling model if initialized with the outputs of a single-slice decoder. A separate formulation of size upscaling is the Parallel Multi-Scale (Reed et al., 2017) ordering where the pixels in an image are doubled at every stage by distinct neural networks and are generated in parallel without sequentiality (3(c)).

## 2.4 Depth Upscaling

Multidimensional upscaling applies upscaling not just in the height and width of the image, but also in the remaining dimension that is channel depth. This is performed in stages such that a network first generates the $d_1$ most significant bits of an image using a conventional or subscale ordering; then a second network generates the next $d_2$ most significant bits of the image conditioned on *all* the $d_1$ bits of the image; and so on to further stages. Using the conventional ordering as basis, the first stage of depth upscaling looks as follows:

$$P(\mathbf{x}^{:\mathbf{d_1}}) = \prod_{h=1}^{H} \prod_{w=1}^{W} \prod_{c}^{\{R,G,B\}} P(\mathbf{x}_{\mathbf{h,w,c}}^{:\mathbf{d_1}} | \mathbf{x}^{:\mathbf{d_1}}_{<}) \tag{3}$$

Next, a second stage of depth upscaling has the following form, conditioned on the first $d_1$ bits of each channel:

$$P(\mathbf{x}^{\mathbf{d_1:d_2}}) = \prod_{h=1}^{H} \prod_{w=1}^{W} \prod_{c}^{\{R,G,B\}} P(\mathbf{x}_{\mathbf{h,w,c}}^{\mathbf{d_1:d_2}} | \mathbf{x}^{\mathbf{d_1:d_2}}_{<}, \mathbf{x}^{:\mathbf{d_1}}) \tag{4}$$

We do not share weights among the networks at different stages of depth upscaling. We note that in depth upscaling bits of lower significance are only generated when the more significants bits at all positions have been generated in a previous stage. Just like for size upscaling from the previous

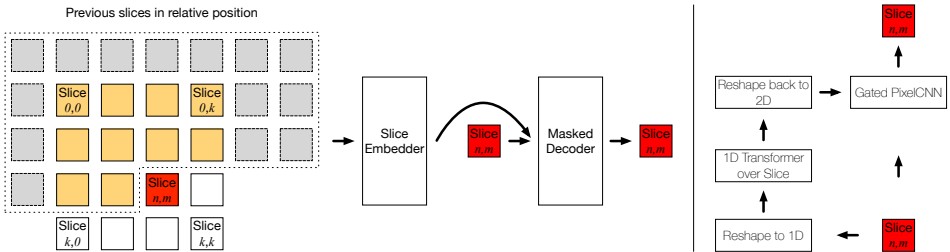

Figure 4: (a) The architecture of a Subscale Pixel Network, with a conditional and a decoding part. (b) Scheme of the parts in the decoder itself.

section, the goal of multidimensional upscaling is to let the model focus on visually salient bits of an image unaffected by less salient and less predictable bits of the image. Depth upscaling is related to the method underlying the Grayscale PixelCNN that models 4-bit greyscale images subsampled from colored images Kolesnikov & Lampert (2016a).

# 3 ARCHITECTURE

## 3.1 CHALLENGES IN TRAINING CONVENTIONAL DECODERS ON LARGE IMAGES

Using a conventional generation ordering, models such as PixelRNN, PixelCNN (van den Oord et al., 2016b) and Image Transformer (Parmar et al., 2018) construct a representation of the generated context for each dimension of each pixel. Existing AR approaches inherently require an amount of computation and memory that is superlinear in the number of pixels. In particular, the quadratic memory requirements of self-attention become severely limiting for images larger than $32 \times 32$ and intractable in practice for the 196,608 distinct positions we consider in a $256 \times 256$ colour image.

Mitigating the memory requirements and computational requirements of encoding the dependencies amongst so many variables often comes at the expense of global context. Modeling choices such as cropping images within the decoders (Kalchbrenner et al., 2016) or performing self-attention over local neighborhoods (Parmar et al., 2018) neglect global dependencies, while model parallelism, though technically feasible with the joint use of a very large number of accelerators, does not overcome the challenges in learning the global structure.

## 3.2 SUBSCALE PIXEL NETWORK

To address these challenges, we devise the Subscale Pixel Network (SPN), an architecture that embodies the subscale ordering from Section 2.2. For an image of size $H \times W \times 3 \times D$, where $D$ is the number of bits used for the current generation stage, one first chooses a scaling factor $S$ and obtains the $S^2$ slices of the original image of size $H/S \times W/S \times 3 \times D$. We use $H = W = 256$ and $S = 8$ as well as $H = W = 128$ and $S = 4$, for the larger images we process, so that the slices have size $32 \times 32 \times 3 \times D$. This scheme of choosing $S$ such that slices are always $32 \times 32$ renders the memory and computation requirements effectively constant as the true image size $H \times W$ changes.

The SPN architecture is composed of two parts: an embedding part for slices at preceding meta-positions that conditions the decoder for the current slice that is being generated (Figure 4 (a)). The embedding part is a convolutional neural network with residual blocks that takes as input preceding slices that are concatenated along the *depth* dimension. One detail is the way the slices are ordered along the channel dimension when concatenated. As illustrated in Figure 4 (a), empty padding slices are used to preserve *relative* meta-positions of each preceding slice with respect to the current target slice. For example, the slice above any target slice in the two-dimensional meta-grid is always aligned in the same position along the depth axis in the input. This achieves equivariance in the embedding architecture with respect to the $(i, j)$ offset of a slice. The padding slices also ensure that the depth of the input slice tensor remains the same for all target slices. In addition to the slice tensor, the embedding part also receives as input the meta-position of the target slice as an embedding of 8 units tiled spatially across the slice tensor. The pixel intensity values are also embedded as one-hot indices of size 8. The context embedding network passes its input through a series of self-attention layers

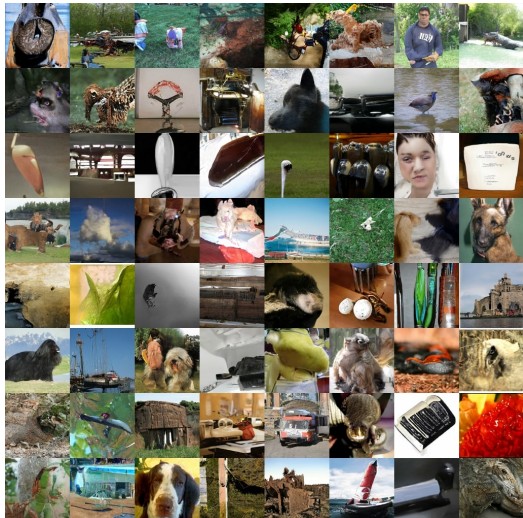 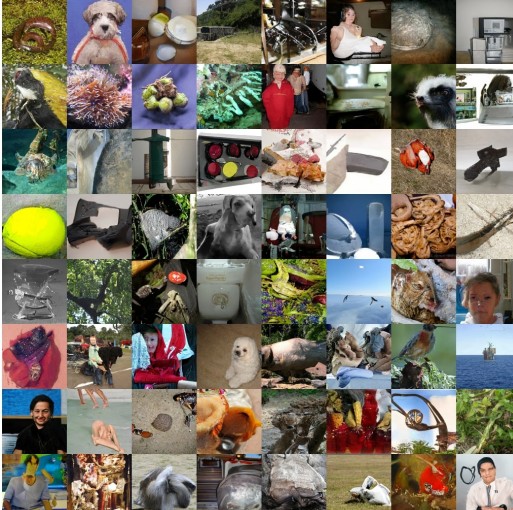

Figure 5: Left: 8-bit 128x128 RGB ImageNet samples from SPN with depth upscaling only. Right: 8-bit 128x128 RGB ImageNet samples from SPN with full-blown Multidimensional Upscaling. Temperature is 0.99 for the initial 32x32 sub-image and otherwise 1.0

and then a series of residual blocks, finally emitting a slice-sized feature map **s** that summarizes the context for the decoder.

The decoder takes as input the encoded slice tensor **s** in a position-preserving manner: each position in the target slice is given as input the encoded representations of pixels at that same position in the preceding slices. In addition it processes the target slice in the raster-scan order. The decoder that we use is a hybrid architecture combining masked convolution and self-attention (Chen et al., 2017). We employ an initial 1D self-attention network Vaswani et al. (2017) that is used to gather the entire available context in the slice (see Figure 4(b)). The slice is reshaped into a 1D tensor before it is given as input to masked 1D self-attention layers; the masking is performed over the previous pixels only (as opposed to over the current RGB channels) in the self-attention layers. Then the output of the layers is reshaped back into a 2D tensor, concatenated depth-wise with the output of the slice embedding network, and given as conditioning input to a Gated PixelCNN as in Equation 5 in van den Oord et al. (2016c). The PixelCNN network models the target slice with full masking over pixels and channel dimensions. We can see how memory requirements are significantly lower - up to $S^2 = 64\times$ lower with $S = 8$ - due to the smaller *spatial* size of the slices and their compact concatenation along the channel dimension of the input tensor. Due to this structure, the entire previously generated context is captured at each position of the decoder.

### 3.3 LEARNING

The log-likelihood derived from equation 2 decomposes as a sum over slices. An unbiased estimator of the log-loss is obtained by uniformly sampling a choice of target slice and evaluating its log-probability conditioned upon previous slices as depicted in Figure 4 (a). We perform maximum likelihood learning by doing stochastic gradient descent on this Monte Carlo estimate, with all gradients computed by backpropagation.

### 3.4 MULTIDIMENSIONAL UPSCALING WITH SPN

As seen in Section 2.3, the SPN naturally serves as a size-upscaling network when the first slice of the input tensor is initialized with with an externally generated subimage. In our experiments, we ensure that the smaller subimages used for the initialization and those used in the training of the SPN decoder are identical to each other.

Analogously, the SPN can be used to upscale the depth of the channels of an image. The image to be upscaled in depth is itself divided into slices by the subscale method (secion 2) and the slices are then

|  | ImageNet 32x32 | ImageNet 64x64 |
|---|---|---|
| Gated PixelCNN (van den Oord et al., 2016c) | 3.83 | 3.57 |
| Parallel Multiscale (Reed et al., 2017) | 3.95 | 3.70 |
| PixelSNAIL (Chen et al., 2017) | 3.80 | - |
| Image Transformer (Parmar et al., 2018) | **3.77** | - |
| Glow (Kingma & Dhariwal, 2018) | 4.09 | 3.81 |
| Decoder baseline | 3.79 | **3.52** |
| SPN | - | 3.53 |
| SPN + Depth Upscaling | - | 3.53 (0.63, 2.90) |
| SPN + 16x16 slices | 3.85 | - |
| SPN + 8x8 slices | 3.91 | - |

Table 1: Negative Log-likelihood (NLL) scores for Downsampled Imagenet (van den Oord et al., 2016b) in bits/dim. The parenthesized numbers in Depth Upscaling rows indicate NLL for $P(\mathbf{x}^{:\mathbf{d_1}})$ and $P(\mathbf{x}_{\mathbf{h,w,c}}^{\mathbf{d_1:d_2}}|\mathbf{x}^{\mathbf{d_1:d_2}}_<, \mathbf{x}^{:\mathbf{d_1}})$ respectively.

|  | ImageNet 128 x 128 | ImageNet 256 x 256 |
|---|---|---|
| Parallel Multiscale (Reed et al., 2017) | 3.55 | - |
| SPN | **3.08** | **2.97** |
| SPN + Depth-Upscaling | **3.08 (0.46, 2.62)** | 3.01 (0.40, 2.61) |

Table 2: NLL scores for high-resolution Imagenet in bits/dim. The parenthesized numbers in Depth Upscaling rows indicate NLL for $P(\mathbf{x}^{:\mathbf{d_1}})$ and $P(\mathbf{x}_{\mathbf{h,w,c}}^{\mathbf{d_1:d_2}}|\mathbf{x}^{\mathbf{d_1:d_2}}_<, \mathbf{x}^{:\mathbf{d_1}})$ respectively.

concatenated along the channel dimension into a slice tensor for the conditioning image $\mathbf{x}^{:\mathbf{d_1}}$. The latter is then added as a fixed additional input to the embedding part of the SPN in order to model $P(\mathbf{x}_{\mathbf{h,w,c}}^{\mathbf{d_1:d_2}}|\mathbf{x}^{\mathbf{d_1:d_2}}_<, \mathbf{x}^{:\mathbf{d_1}})$. The model of $P(\mathbf{x}^{:\mathbf{d_1}})$ is a normal SPN, but trained on data with low bit depth.

## 4 EXPERIMENTS

We demonstrate experimentally that our model is capable of high fidelity samples at high resolution, producing unconditional CelebA-HQ samples of quality better than the Glow model (Kingma & Dhariwal, 2018) and improving the MLE scores. Furthermore, we show that these results extend to high-resolution ImageNet images, with state-of-the-art log-likelihoods at 128x128 by a large margin and the first benchmark on 256x256 ImageNet. Unconditional samples at these resolutions are characterized by unprecedented global coherence.

Because our networks operate on small images ($32 \times 32$ slices), we can train large networks both in terms of the number of hidden units and in terms of network depth (see Appendix C for details of sizes). The context-embedding network contains 5 convolutional layers and 6-8 self-attention layers depending on the dataset. The masked decoder consists of a PixelCNN with 15 layers in all experiments. The 1D Transformer in the decoder (Figure 4(b)) has between 8 and 10 layers depending on the dataset. See Table 4 for all dataset-specific hyperparameter details.

### 4.1 DOWNSAMPLED IMAGENET AT $32 \times 32$ AND $64 \times 64$

We first benchmark the performance of our hybrid decoder alone (i.e. no subscaling, Figure 4(b)) and show that it compares favorably to state of the art models on $32 \times 32$ Downsampled ImageNet (see Table 1). We find that SPN hurts in this low-resolution setting with $S = 2$ and even further with $S = 4$. This is likely because the size of the resulting image slices becomes very small and the image coarse grained.

On $64 \times 64$ Downsampled ImageNet, we achieve a state of the art log-likelihood of 3.52 bits/dim. We hypothesize that PixelSNAIL would achieve a similar score, but results at this resolution were not reported in Chen et al. (2017). At this resolution, SPN scores similarly with 3.53 bits/dim. The improvement over Glow in the 5-bit setting is very significant (Table 3).

|  | ImageNet 64 x 64 (5bit) | CelebA-HQ 256 x 256 (5bit) |
|---|---|---|
| Glow (Kingma & Dhariwal, 2018) | 1.76 | 1.03 |
| SPN | **1.41** | **0.61** |

Table 3: Negative Log-likelihood scores for 5-bit datasets in bits/dim.

## 4.2 IMAGENET AT $128 \times 128$

For these experiments we use the standard ILSVRC Imagenet dataset (Kolesnikov & Lampert, 2016b) resized with Tensorflow's resize_area function. Parallel Multiscale PixelCNN (Reed et al., 2017) is the only model in the literature which reports log-likelihood on $128 \times 128$ ImageNet. SPN improves the log-likelihood over this model from 3.55 bits/dim to 3.08 bits/dim (see Table 2). Figure 5 gives $128 \times 128$ 8-bit ImageNet samples for both the setting of depth upscaling only and of complete multidimensional upscaling. These settings do not affect the NLL, but the samples with depth upscaling show significant semantic coherence that is usually lacking in samples without upscaling. In addition, multidimensional upscaling seems to increase the overall rate of success of the samples. Additional intermediate ImageNet samples can be seen in Figures 10, 11 and 12 in the Appendix.

## 4.3 CELEBAHQ

At $256 \times 256$ we can produce high-fidelity samples of celebrity faces from the CelebAHQ dataset. The quality compares favorably to the samples of other models such as Glow and GANs (Karras et al., 2017). We show in Table 3 that the achieved MLE scores are a significant improvement over previously reported scores. Figure 6 showcases some samples for 8-bit CelebAHQ-256. Figure 7 in the Appendix includes 5-bit samples, Figure 8 includes 3-bit samples while Figure 9 includes 3-bit samples with the temperature of the output distribution set to 0.95.

## 5 CONCLUSION

The problem of whether it is possible to learn the distribution of complex natural images and attain high sample fidelity has been a long-standing one in the tradition of generative models. The SPN and Multidimensional Upscaling model that we introduce accomplishes a large step towards solving this problem, by attaining both state-of-the-art MLE scores on large-scale images from complex domains such as CelebAHQ-256 and ImageNet-128 and by being able to generate high fidelity full 8-bit samples from the resulting learnt distributions without alterations to the sampling process (via e.g. heavy modifications of the temperature of the output distribution). The generated samples show an unprecedented amount of semantic coherence and exactness of details even at the large scale size of full 8-bit $128 \times 128$ and $256 \times 256$ images.

## 6 ACKNOWLEDGEMENTS

We would like to thank Alex Graves, Karen Simonyan, Aaron van den Oord, Tim Harley, Sander Dieleman, Tim Salimans, Lasse Espeholt, Ali Razavi, Jeffrey De Fauw and Andy Brock for insightful discussions. In particular we wish to thank Andriy Mnih for formative ideas about autoregressivity in the bit-depth of an image.

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

## APPENDIX A

In some cases for purposes of analysis the entropy of the softmax output distributions has been artificially reduced via a "temperature" divisor on the predicted logits. When we say the temperature is 0.95, we mean that the logits of a trained model have been divided by this constant at sampling time.

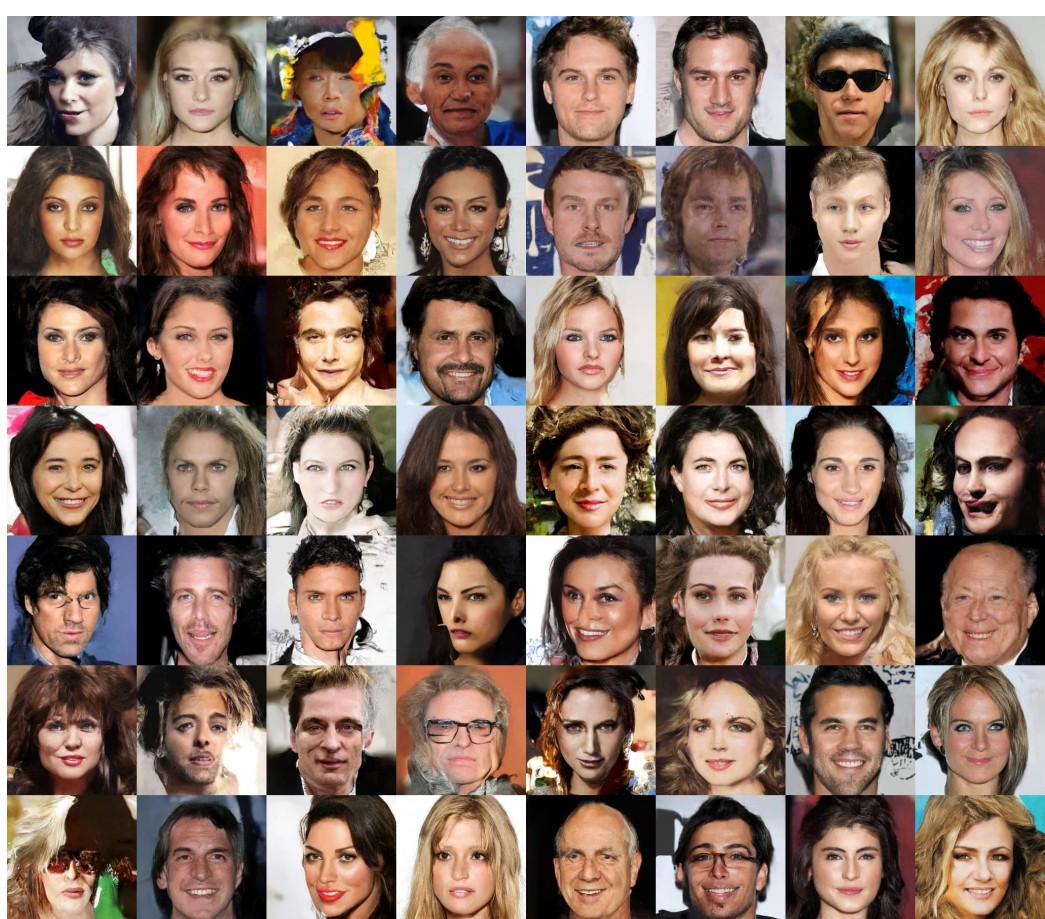

Figure 6: 8-bit 256x256 RGB CelebA-HQ samples from SPN with Depth-Upscaling. Temperature is 0.99 for the low-bit-depth image and otherwise 1.0

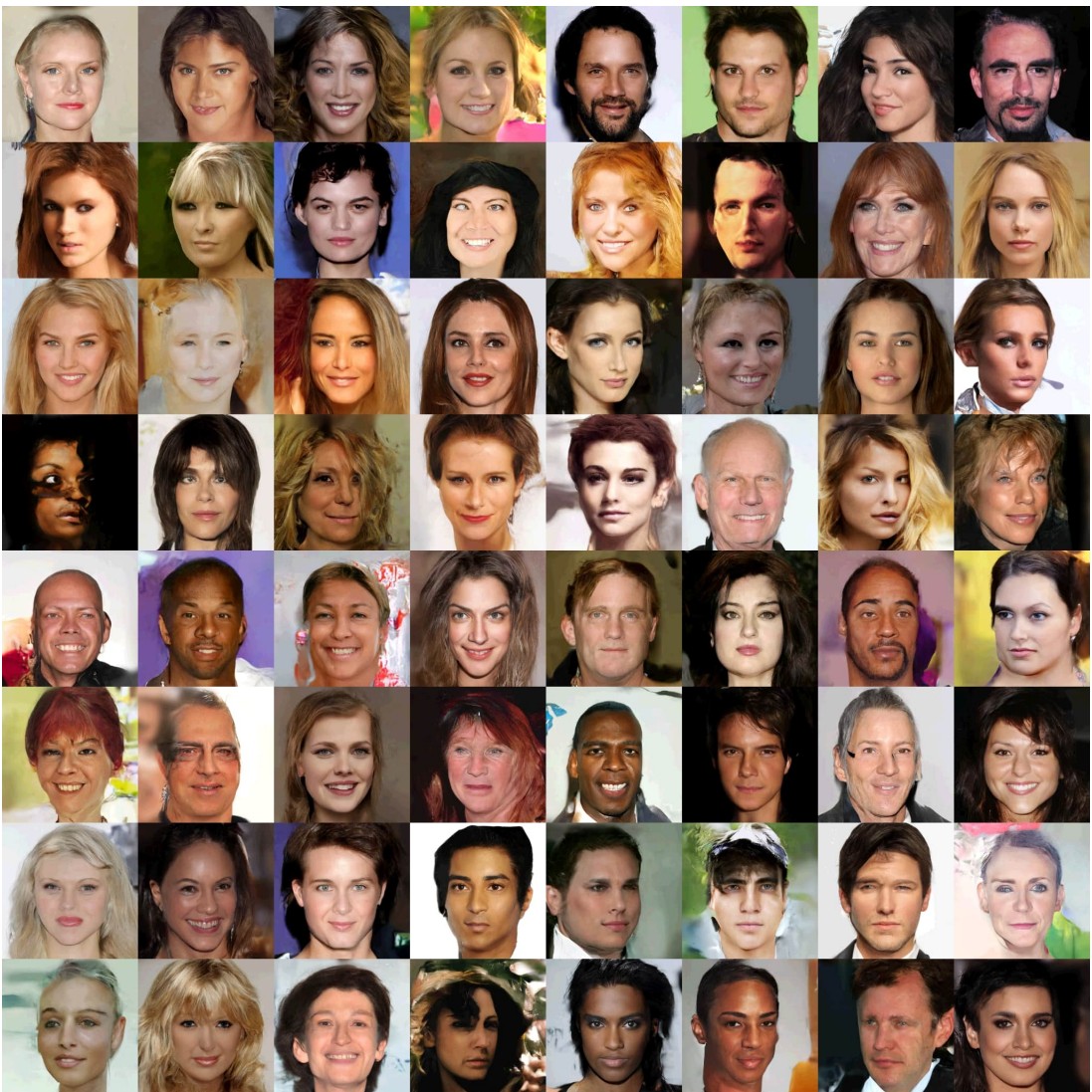

Figure 7: 256x256 CelebA-HQ 5bit samples from SPN

## APPENDIX B

Please see Table 4 for all the detailed hyperparameters.

## APPENDIX C

Our experiments operate at a fairly large scale in terms of both the amount of compute used and the size of the networks. We proportionately increase the batch size so that the number of pixels in a batch is not affected by the subscaling. These large batch sizes (a maximum of 2048) are achieved by increasing the degree of data parallelism by running on Google Cloud TPU pods (Jouppi et al., 2017). For Imagenet 32 we used 64 tensorcores. For ImageNet 64, 128 and 256, we use 128 tensorcores. The fast interconnect between these devices affords much faster synchronous gradient computation than would be possible using the same number of GPUs. When overfitting is a problem, as in small datasets like CelebA-HQ, we rather decrease the batch size and use a lower number of 32 tensorcores.

Our SPN architectures have between ∼50M and ∼250M parameters depending on the dataset. See Table 4 for the number of parameters in the SPN architecture for each dataset. Depth-upscaling

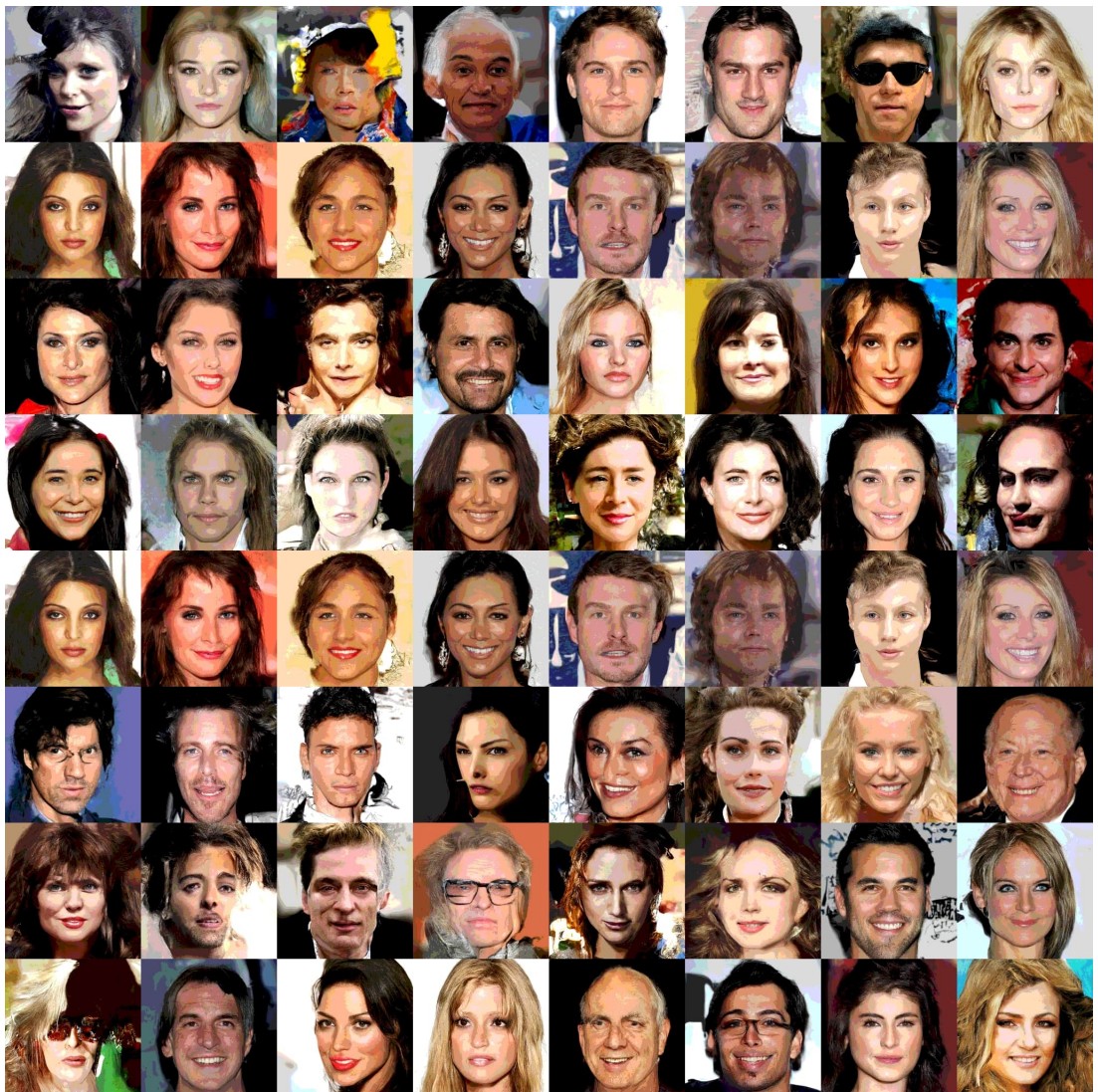

Figure 8: 256x256 CelebA-HQ 3bit samples from SPN

doubles the number of parameters due to using two separate networks with untied weights. Size-upscaling adds more parameters still for the separate decoder-only network which models the first slice as seen in Figure 3 (g). Thus the maximal number of parameters used to generate a sample in the paper occurs in the multidimensional upscaling setting for ImageNet 128, where the total parameter count reaches ∼650M (the decoder-only network used to model the first slice has ∼150M parameters).

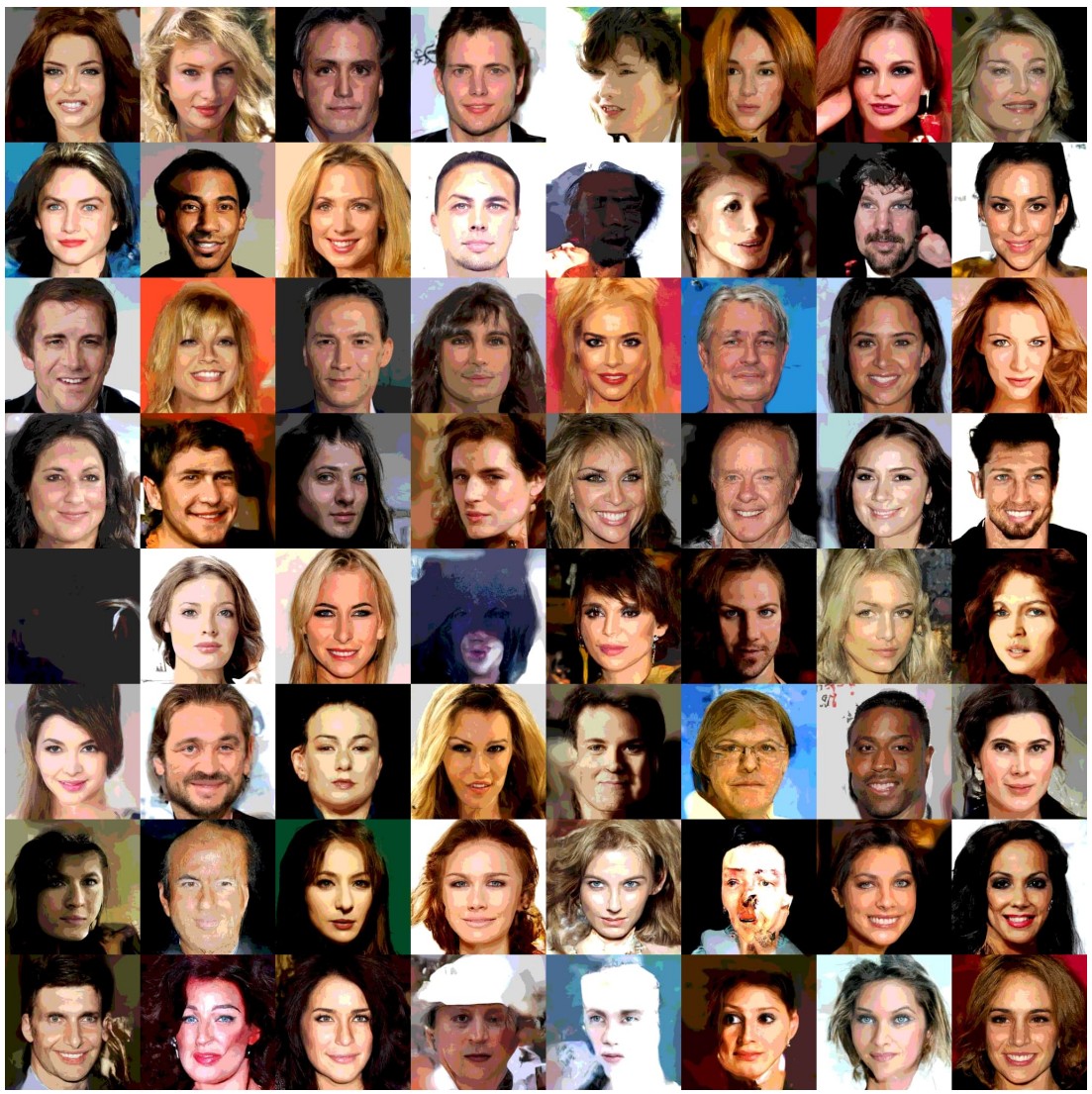

Figure 9: 256x256 CelebA-HQ 3bit samples from SPN with temperature 0.95

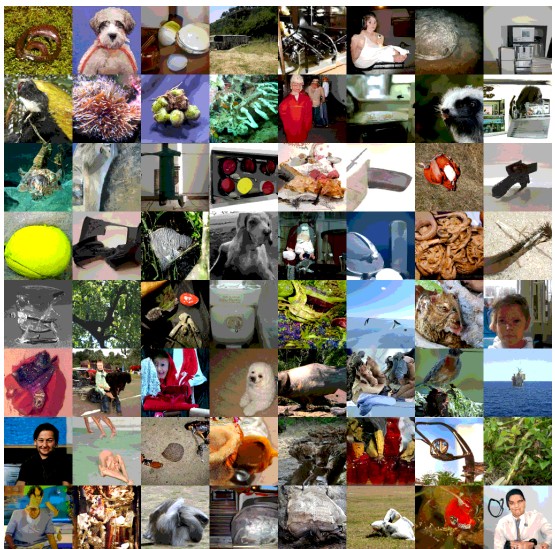

Figure 10: 128x128 ImageNet 3bit; upscaled 32x32 slices

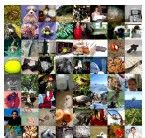

Figure 11: 128x128 ImageNet 3bit samples from model trained on 32x32 slices

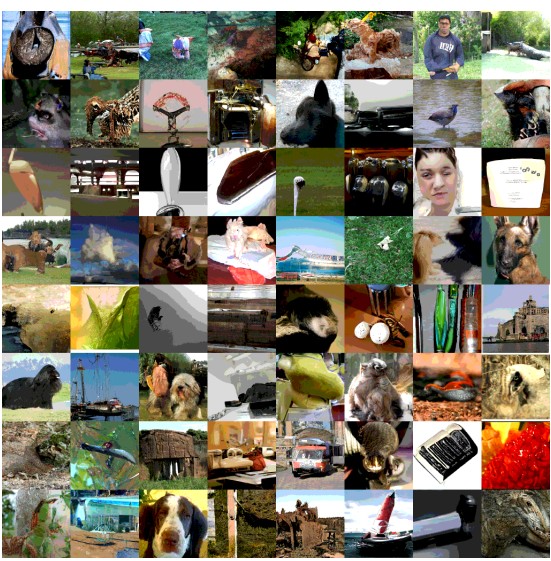

Figure 12: 128x128 ImageNet 3bit samples from SPN

| | ImageNet (32, 64, 128, 256) | CelebA HQ |
|---|---|---|
| **Optimization** | | |
| batch size | (1024, 2048, 2048, 2048) | 256 |
| learning rate | (sched, sched, 1e-5, 1e-5) | 5e-5 |
| rmsprop momentum | 0.9 | 0.9 |
| rmsprop decay | 0.95 | 0.95 |
| rmsprop epsilon | 1e-8 | 1e-8 |
| polyak decay | 0.9999 | 0.9999 |
| **Decoder** | | |
| PixelCNN layers | 15 | 15 |
| PixelCNN conv channels | (256, 256, 384, 384) | 128 |
| PixelCNN residual channels | 1280 | 1280 |
| PixelCNN nonlinearity | gated | gated |
| PixelCNN filter size | 3 | 3 |
| masked self-attention layers | 8 | 5 |
| attention heads | 10 | 5 |
| attention channels | 128 | 128 |
| attention ffn layer | "parameter_attention" | "parameter_attention" |
| **Slice Embedder** | | |
| conv layers | 5 | 5 |
| conv filter size | 3 | 3 |
| conv channels | 256 | 256 |
| residual channels | 1024 | 1024 |
| nonlinearity | relu | relu |
| self-attention layers | (6, 6, 8, 8) | 6 |
| attention heads | (4, 4, 8, 8) | 4 |
| attention channels | (64, 64, 128, 128) | 64 |
| attention ffn layer | "parameter_attention" | "parameter_attention" |
| **Number of parameters** | | |
| | ( $\sim$ 150M, $\sim$ 150M, $\sim$ 250M, $\sim$ 250M) | $\sim$ 50M |

Table 4: SPN Hyperparameters. A learning rate marked "sched" utilizes a piecewise-constant schedule starting at 1e-4, and decreasing to 3e-5 and finally 1e-5 at training steps 50k and 100k respectively. The "attention" parameters listed are configurable hyperparameters of the open source Transformer implementation in Vaswani et al. (2018). The parameter "residual channels" refers to the number of hidden units in residual convolution layers within the Slice Embedder or PixelCNN networks.

