# OpenReview forum: "GENERATING HIGH FIDELITY IMAGES WITH SUBSCALE PIXEL NETWORKS AND MULTIDIMENSIONAL UPSCALING"
_ICLR.cc/2019/Conference_

### Official Review · AnonReviewer3 · 2018-11-01
**A new version of PixelCNN-based model for HQ images**

**Rating:** 9
**Confidence:** 3

**Review:**

General:
The paper tackles a problem of learning long-range dependencies in images in order to obtain high fidelity images. The authors propose to use a specific architecture that utilizes three main components: (i) a decoder for sliced small images, (ii) a size-upscaling decoder for large image generation, (iii) a depth-upscaling decoder for generating high-res image. The main idea of the approach is slicing a high-res original image and a new factorization of the joint distribution over pixels. In this model various well-known blocks are used like 1D Transformer and Gated PixelCNN. The obtained results are impressive, the generated images are large and contain realistic details.

In my opinion the paper would be interesting for the ICLR audience.

Pros:
+ The paper is very technical but well-written.
+ The obtained results constitute new state-of-the-art on HQ image datasets.
+ Modeling long-range dependencies among pixels is definitely one of the most important topics in image modeling. The proposed approach is a very interesting step towards this direction.

Cons:
- The authors claim that the proposed approach is more memory efficient than other methods. However, I wonder how many parameters the proposed approach requires comparing to others. It would be highly beneficial to have an additional column in Table 1 that would contain number of parameters for each model.
- All samples are take either at an extremely high temperature (i.e., 0.99) or at the temperature equal 1. How do the samples look for smaller temperatures? Sampling at very high temperature is a nice trick for generating nicely looking images, however, it could hide typical problems of generative models (e.g., see Rezende & Viola, “Taming VAEs”, 2018).

--REVISION--
I would like to thank the authors for their response. I highly appreciate their clear explanation of both issues raised by me. I am especially thankful for the second point (about the temperature) because indeed I interpreted it as in the GLOW paper. Since both my concerns have been answered, I decided to raise the final score (+2).

---

> ### Author Response · Authors · 2018-11-14
> **reply to AnonReviewer3**
>
> Thank you for your comments.
>
> --
> - The authors claim that the proposed approach is more memory efficient than other methods. However, I wonder how many parameters the proposed approach requires comparing to others. It would be highly beneficial to have an additional column in Table 1 that would contain number of parameters for each model.
> --
>
> As discussed with AnonReviewer2, we will include a table with the number of parameters for each model. Briefly, the models in the paper have between ~50M params and ~650M params in the most extreme case of full multidimensional upscaling on ImageNet 128.
>
> In the case of 256x256 CelebA-HQ, we use a total of ~100M parameters to produce the depth-upscaled 8bit samples in Figure 5 and ~50M parameters to produce the 5bit samples in Figure 7. Compare this to Glow [1], whose blog post [2] indicates that up to 200M parameters are used for 5bit Celeb-A. Thus we have a ~4x reduction in the number of parameters vs Glow, with decisively improved likelihoods (see Table 3); I think this should address your concern about parameter-efficiency. We also note that autoregressive (and other) models are highly compressible at little to no loss (see e.g. [3]), which makes the absolute number of parameters only an initial, rough measure of parameter efficiency.
>
> --
> - All samples are take either at an extremely high temperature (i.e., 0.99) or at the temperature equal 1. How do the samples look for smaller temperatures? Sampling at very high temperature is a nice trick for generating nicely looking images, however, it could hide typical problems of generative models (e.g., see Rezende & Viola, “Taming VAEs”, 2018).
> --
>
> I believe there is a misunderstanding here. What we call temperature is a division on the logits of the softmax output distribution. Temperature 1.0 in our case means that the distribution of the trained model is used exactly as predicted by the model, with no adjustments or tweaks during sampling time.  *Reducing* the temperature (less than 1.0) is what can hide problems, because it artificially reduces the entropy in the distribution parameterized by the model during sampling time.
>
> As we sample at temperatures of 0.95, 0.99, and 1.0 in the paper, we respectively *slightly*, *barely*, and *do-not-at-all* reduce the entropy in the model's distribution. Thus this concern does not apply and we are actually being comparatively transparent about our model’s samples (note that Glow shows its best samples at temperature 0.7, but that “temperature” has a different operational meaning in that case).
>
> [1] - Kingma et al. https://arxiv.org/abs/1807.03039
> [2] - https://blog.openai.com/glow/
> [3] - Kalchbrenner et al. https://arxiv.org/abs/1802.08435

---

### Official Review · AnonReviewer2 · 2018-11-02
**Solid paper, excellent execution, very important advance in density modeling**

**Rating:** 10
**Confidence:** 5

**Review:**

Summary:
This paper addresses an important problem in density estimation which is to scale the generation to high fidelity images. Till now, there have been no good density modeling results on large images when taken into account large datasets like Imagenet (there have been encouraging results like with Glow, but on 5-bit color intensities and simpler datasets like CelebA). This paper is the first to successfully show convincing Imagenet samples with 128x128 resolution for a likelihood density model, which is hard even for a GAN (only one GAN paper (SAGAN) prior to this conference has managed to show unconditional 128x128 Imagenet samples). The ideas in this paper to pick an ordering scheme at subsampled slices uniformly interleaved in the image and condition slice generation in an autoregressive way is very likely to be adopted/adapted to more high fidelity density modeling like videos. Another important idea in this paper is to do depth upscaling, focusing on salient color intensity bits first (first 3 bits per color channel) before generating the remaining bits. The color intensity dependency structure is also neat: The non-salient bits per channel are conditioned on all previously generated color bits (for all spatial locations). Overall, I think this paper is a huge advance in density modeling, deserves an oral presentation and deserves as much credit as BigGAN, probably more, given that it is doing unconditional generation.

Details:
Major:
-1. Can you point out the total number of parameters in the models? Also would be good to know what hardware accelerators were used. The batch sizes mentioned in the Appendix (2048 for 256x256 Imagenet) are too big and needs TPUs? If TPU pods, which version (how many cores)? If not, I am curious to know how many GPUs were used.
0. I would really like to know the sampling times. The model still generates the image pixel by pixel. Would be good to have a number for future papers to reference this.
1. Any reason why 256x256 Imagenet samples are not included in the paper? Given that you did show 256x256 CelebA samples, sampling time can't be an issue for you to not show Imagenet 256x256. So, it would be nice to include them. I don't think any paper so far has shown good 256x256 unconditional samples. So showing this will make the paper even stronger.
2. Until now I have seen no good 64x64 Imagenet samples from a density model. PixelRNN samples are funky (colorful but no global structure). So I am curious if this model can get that. It may be the case that it doesn't, given that subscale ordering didn't really help on 32x32.  It would be nice to see both 5-bit and 8-bit, and for 8-bit, both the versions: with and without depth upscaling.
3. I didn't quite understand the architecture in slice encoding (Sec 3.2).  Especially the part about using a residual block convnet to encode the previous slices with padding, and to preserve relative meta-position of the slices. The part I get is that you concatenate the 32x32 slices along the channel dimension, with padded slices. I also get that padding is necessary to have the same channel dimension for any intermediate slice. Not sure if I see the whole point of preserving ordering. Isn't it just normal padding -> space to depth in a structured block-wise fashion?
4. Can you clarify how you condition the self-attention + Gated PixelCNN block on the previous slice embedding you get out of the above convnet? There are two embeddings passed in if I understand correctly: (1) All previous slices, (2) Tiled meta-position of current slice.  It is not clear to me how the conditioning is done for the transformer pixelcnn on this auxiliary embedding. The way you condition matters a lot for good performance, so it would be helpful for people to replicate your results if you provide all details.
5. I also don't understand the depth upscaling architecture completely. Could you provide a diagram clarifying how the conditioning is done there given that you have access to all pixels' salient bits now and not just meta-positions prior to this slice?
6. It is really cool that you don't lose out in bits/dim after depth upscaling that much. If you take Grayscale PixelCNN (pointed out in the anonymous comment), the bits/dim isn't as good as PixelCNN though samples are more structured. There is 0.04 b.p.d  difference in 256x256, but no difference in 128x128. Would be nice to explain this when you add the citation.
7. The architecture in the Appendix can be improved. It is hard to understand the notations. What are residual channels, attention channels, attention ffn layer, "parameter attention", conv channels?

Minor:
Typo: unpredented --> unprecedented

---

> ### Comment · AnonReviewer2 · 2018-11-14
> **More details on the 128x128 and 256x256 Imagenet benchmarks**
>
> Request the authors to provide details for the train/val split for Imagenet 128x128 and Imagenet 256x256 density estimation benchmarks. I wasn't able to find the details in Reed et al (https://arxiv.org/pdf/1703.03664.pdf). Would be ideal if the authors released the splits as done for 32x32 and 64x64 from PixelRNN in http://image-net.org/small/download.php to encourage and be useful for more people to push on this benchmark.

---

> > ### Author Response · Authors · 2018-11-14
> > **re: More details on the 128x128 and 256x256 Imagenet benchmarks**
> >
> > Thanks for your thoroughness, AnonReviewer2.
> >
> > The ImageNet dataset we use for 128x128 and 256x256 generation is the standard ILSVRC [1] benchmark used by classification models. We report final numbers on the official validation set consisting of 50k examples. We hold out 10k examples from the official training set for cross-validation, and train on the remaining 1271167 points.
> >
> > I don’t believe a separate release is necessary here as the data is freely available [2] and our downsampling scheme is easily reproducible (simply tf.resize_area).
> >
> > We can be more explicit about this split in the experiment section.
> >
> > [1] - Russakovsky et al. ImageNet Large Scale Visual Recognition Challenge. IJCV, 2015.
> > [2] - http://www.image-net.org/challenges/LSVRC/2014/

---

> > > ### Comment · AnonReviewer2 · 2018-11-14
> > > **thanks for the clarification!**
> > >
> > > Thanks for the clarification. Agreed a separate release isn't needed. It wasn't clear if you were using the same split as Reed et al doesn't talk about it.

---

> ### Author Response · Authors · 2018-11-14
> **reply to AnonReviewer2 (1/2)**
>
> Thanks for your thorough review. Addressing your comments will improve the paper.
>
> --
> -1. Can you point out the total number of parameters in the models?
> --
>
> Depending on the dataset, each SPN network has between ~50M (CelebA) and ~250M parameters (ImageNet 128/256). ImageNet64 uses ~150M weights. With depth upscaling, two separate SPNs with non-shared weights model P(3bit) and P(rest given 3bit) respectively, doubling the number of parameters. With explicit size upscaling for ImageNet128, there is a third network (decoder-only) with ~150M parameters which generates the first 3 bits of the first slice. So the maximal number of parameters used to generate a sample in the paper is full multidimensional upscaling on ImageNet 128, where the total parameter count reaches ~650M. We will include the number of parameters for each model in the table, as requested.
>
> --
> -1. Also would be good to know what hardware accelerators were used. The batch sizes mentioned in the Appendix (2048 for 256x256 Imagenet) are too big and needs TPUs? If TPU pods, which version (how many cores)?
> --
>
> To reach batch size 2048 we used 256 TPUv3 cores. We will clarify this in the paper.
>
> --
> 0. I would really like to know the sampling times. The model still generates the image pixel by pixel. Would be good to have a number for future papers to reference this.
> --
>
> Our current implementation performs only naive sampling, where the outputs of the decoder are recomputed for all positions in a slice to generate each sample. This is convenient, but time-consuming and allows to only rarely inspect the samples coming from our model. The techniques for speeding-up AR inference - such as caching of states, low-level custom implementation, sparsification and multi-output generation [1] - are equally applicable to SPNs and would make sampling reasonably fast; on the order of a handful of seconds for a 256 x 256 x 3 image.
>
> --
> 1. Any reason why 256x256 Imagenet samples are not included in the paper? Given that you did show 256x256 CelebA samples, sampling time can't be an issue for you to not show Imagenet 256x256. So, it would be nice to include them. I don't think any paper so far has shown good 256x256 unconditional samples. So showing this will make the paper even stronger.
> --
>
> Thanks! We’ll aim to adding 64 x 64 and 256 x 256 samples in our revision.
>
> --
> 2. Until now I have seen no good 64x64 Imagenet samples from a density model. PixelRNN samples are funky (colorful but no global structure). So I am curious if this model can get that. It may be the case that it doesn't, given that subscale ordering didn't really help on 32x32.  It would be nice to see both 5-bit and 8-bit, and for 8-bit, both the versions: with and without depth upscaling.
> --
>
> The 64x64 samples look much better with SPNs. We will aim at including some of the variants that you ask for in our revision.
>
> --
> 3. I didn't quite understand the architecture in slice encoding (Sec 3.2).  Especially the part about using a residual block convnet to encode the previous slices with padding, and to preserve relative meta-position of the slices. The part I get is that you concatenate the 32x32 slices along the channel dimension, with padded slices. I also get that padding is necessary to have the same channel dimension for any intermediate slice. Not sure if I see the whole point of preserving ordering. Isn't it just normal padding -> space to depth in a structured block-wise fashion?
> --
>
> It’s like a meta-convolution: the relative ordering ensures that slices are embedded with weights that depend on the relative 2d distance to the slice that is being generated. Suppose we are predicting the target slice at meta-position (i,j), so that previous slices in the 2d ordering are presented to the slice embedder. For any previous slice (m,n), the weights applied to it are a function of the offset (i-m,j-n), as opposed to their absolute positions (m,n). We will add this clarification to the paper.

---

> ### Author Response · Authors · 2018-11-14
> **Reply to AnonReviewer2 (2/2)**
>
> --
> 4. Can you clarify how you condition the self-attention + Gated PixelCNN block on the previous slice embedding you get out of the above convnet? There are two embeddings passed in if I understand correctly: (1) All previous slices, (2) Tiled meta-position of current slice.  It is not clear to me how the conditioning is done for the transformer pixelcnn on this auxiliary embedding. The way you condition matters a lot for good performance, so it would be helpful for people to replicate your results if you provide all details.
> --
>
> The output of the slice embedder -- that receives as input all previous slices and the tiled meta-position of the current slice -- is concatenated channel-wise with the 2D-reshaped output of the masked 1D transformer (which in turn receives as input only the current target slice). The resulting concatenated tensor conditions the PixelCNN decoder like $s$ in equation (5) of the Conditional PixelCNN paper [2]. I.e. the tensor $s$ maps, via 1x1 convolutions, to units which bias the masked convolution output for each layer in PixelCNN. The number of hidden units in this pathway is what is referred to as "decoder residual channels" in Appendix B. We will add this description to Section 3.2
>
> --
> 5. I also don't understand the depth upscaling architecture completely. Could you provide a diagram clarifying how the conditioning is done there given that you have access to all pixels' salient bits now and not just meta-positions prior to this slice?
> --
>
> The SPN which models the low-bit-depth image is identical to the exposition in section 3.2, except that the data it operates on has only 3 bits of depth. As we mention in section 3.4, the depth-upscaling SPN achieves its conditioning by concatenating (again channelwise) the full low-bit-depth image, organised into its constituent slices, to the rest of the slice-embedder's inputs. So no matter which target slice is being modelled (for the finest 5 bits), all slices of the 3bit data can be seen by the slice embedder when it produces context for the fine bits of a target slice. We will further clarify it and see how to add a diagram for this.
>
> --
> 6. It is really cool that you don't lose out in bits/dim after depth upscaling that much. If you take Grayscale PixelCNN (pointed out in the anonymous comment), the bits/dim isn't as good as PixelCNN though samples are more structured. There is 0.04 b.p.d  difference in 256x256, but no difference in 128x128. Would be nice to explain this when you add the citation.
> --
>
> Thanks for the observation, we will note this. The ordering in the SPN, considering both subscaling and upscaling, is indeed quite different from the vanilla ordering and it’s nice to see that the NLL values are negligibly affected.
>
> --
> 7. The architecture in the Appendix can be improved. It is hard to understand the notations. What are residual channels, attention channels, attention ffn layer, "parameter attention", conv channels?
> --
>
> Thanks for bringing this to our attention. We will add the figures/explanations discussed and reference this hyperparameter table so that it's all clear. The attention parameters listed are configurable hyperparameters of the open source Transformer implementation in tensor2tensor [3] on github.
>
>
> [1] - Kalchbrenner et al. https://arxiv.org/abs/1802.08435
> [2] - Oord et al. https://arxiv.org/abs/1606.05328
> [3] - https://github.com/tensorflow/tensor2tensor
>
> And we'll fix that typo too.

---

### Official Review · AnonReviewer1 · 2018-11-05
**Sound incremental advance**

**Rating:** 7
**Confidence:** 3

**Review:**

Authors propose a decoder arquitecture model named Subscale Pixel Network. It is meant to generate overall images as image slice sequences with memory and computation economy by using a Multidimensional Upscaling method.
The paper is fairly well written and structured, and it seems technically sound.
Experiments are convincing.
Some minor issues:
Figure 2 is not referenced anywhere in the main text.
Figure 5 is referenced in the main text after figure 6.
Even if intuitively understandable, all parameters in equations should be explicitly described (e.g., h,w,H,W in eq.1)

---

> ### Author Response · Authors · 2018-11-14
> **reply to AnonReviewer1**
>
> Thank you for the detailed feedback. In the next revision, we will make height, width, and channel indices in equation 1 explicit and make a thorough sweep over the rest of the equations to check for any other undefined parameters.
>
> We will ensure that all figures are referenced, and in the correct order.

---

### Public Comment · (anonymous) · 2018-09-30
**Great results but very relevant work discussion missing?**

https://arxiv.org/pdf/1612.08185 also proposed both low-resolution and sub-pixel color modelling.

---

> ### Author Response · Authors · 2018-09-30
> **Thanks for the reference. Low resolution (small height/width) modeling goes back to Multi-Scale PixelRNN.**
>
> Thanks for the reference - we will add the citation in the context of depth upscaling. Size upscaling in AR models goes back to at least the PixelRNN paper (van den Oord et al, 2016, see Multi-Scale section).
>
> Some differences:
> - Depth upscaling here is done by taking the most significant bits of each channel separately, as opposed to globally across the three channels as in Grayscale PixelCNN.
> - Multidimensional Upscaling used here combines both size and depth upscaling.

---

> > ### Public Comment · (anonymous) · 2018-12-14
> > **Related work**
> >
> > https://arxiv.org/abs/1109.4389 seems to be another relevant reference for AR models using multiple scales

---

### Author Response · Authors · 2018-11-26
**New revision is now uploaded**

To our reviewers,

Please find our latest revision uploaded. We believe it addresses most of the comments including:
- Detailing the number of parameters for each architecture (Table 4)
- Clarifying exactly how the slice embedder conditions the decoder
- Clarifying depth-upscaling with SPN
- Including information about the use of TPU (Appendix C)
- Supplying details about the nature of temperature adjustments during sampling
- Adding references
- Various writing improvements

We are also currently running our setup for the 64x64 and 256x256 samples with the intent to include them shortly in our revision.

We kindly thank our reviewers for their insightful comments which have substantially improved the exposition.

---

### Public Comment · ~Kun_Xu1 · 2018-12-06
**A concern about depth-upscaling**

Dear authors:

Thank you for your really interesting and impressive ideas. The idea is really amazing and experimental results are sound. Generating 256x256 imagenet images in the auto-regressive manner is really difficult and your paper gives a really solid solution.

However, about this paper, I have a concern about the depth-upscaling part. In your experimental results, the bits/dim of SPN and SPN+ depth-upscaling is of no difference for most datasets, and sometiems the SPN+depth-upscaling even performs poorly compared to simply SPN. However, with the depth-upscaling, the sampling time is doubled: every dimension should be sampled twice compared SPN. Can you give more explanations on the benefits of depth-upscaling? Do we really need it given the really impressive results of SPN?

Anyway, this is a really solid paper and congratulations.

---

### Meta-Review · Area_Chair1 · 2018-12-10
**metareview: significant progress on autoregressive models for image generation**

**Confidence:** 4
**Recommendation:** Accept (Oral)

**Metareview:**

All reviewers recommend acceptance, with two reviewers in agreement that the results represent a significant advance for autoregressive generative models. The AC concurs.